# OpenReview forum: "SpeechMedAssist: Efficiently and Effectively Adapting Speech Language Model for Medical Consultation"
_ICLR.cc/2026/Conference — ICLR 2026 Conference Withdrawn Submission_

### Official Review · Reviewer_TkSz · 2025-10-31

**Soundness:** 2
**Presentation:** 2
**Contribution:** 2
**Rating:** 4
**Confidence:** 4

**Summary:**

Current medical consultation research primarily focuses on fine-tuning text-based large language models and does not support spoken interaction. To address this gap, this paper propose SpeechMedAssist, a speech language model natively capable of multi-turn spoken interactions between patients and clinicians. To overcome the scarcity of medical speech data, they introduce a two-stage alignment strategy. Experimental results show that SpeechMedAssist outperforms baseline models on the authors’ newly constructed speech-based medical benchmark.

**Strengths:**

1. The authors introduced a multi-turn spoken medical dialogue dataset, SpeechMedDataset, containing 198k samples, and used it to fine-tune a speech large language model, thereby obtaining a model with enhanced capabilities for spoken medical question answering.

2. Through careful experimental analysis, the authors demonstrated the effectiveness of their proposed dataset.

**Weaknesses:**

1. The primary contribution of this work lies in data construction—specifically, the introduction of SpeechMedDataset, a multi-turn spoken medical dialogue dataset. However, the model architecture, training strategy, and evaluation protocol largely follow those of Llama-Omni and Llama-Omni2, demonstrating limited novelty in methodological design.

2. In Table 2, the authors do not include a direct comparison between:\
(a) the base text-only model (Qwen2.5-Instruct-7B) and its performance on medical tasks,\
(b) a cascaded pipeline using Qwen2.5-Instruct-7B, and\
(c) the performance of the text LLM after Stage 1 fine-tuning.\
Such comparisons would better illustrate the performance gap between text and speech modalities and more convincingly demonstrate the advantages of their end-to-end approach over cascaded systems.

3. It is puzzling that SMA-Stage1, after fine-tuning on LLaMA-Omni2, underperforms the original Llama-Omni2 in multi-turn medical question answering. The authors should provide an analysis or ablation study to explain this performance degradation—e.g., whether it stems from domain shift.

**Questions:**

Please refer to the Weaknesses.

---

> ### Author Response · Authors · 2025-11-26
> **Author Response (1/n)**
>
> Dear Reviewer, before addressing the valuable weaknesses you pointed out, we believe a brief clarification is necessary. After carefully reading your review comments, we think there may be a misunderstanding of our work. It appears that you may have interpreted our method as either (1) following LLaMA-Omni2 and training a Speech Encoder + Qwen2.5-7B-Instruct model from scratch on our constructed SpeechMedDataset (as mentioned in Weakness 1&2), or (2) directly fine-tuning LLaMA-Omni2 using SpeechMedDataset (as mentioned in Weakness 3). However, neither of these is the way of our work.
>
> Our goal is to **efficiently adapt a pretrained general-purpose SpeechLM to a vertical domain such as healthcare**. The key challenge we face is the scarcity of in-domain speech data, which makes it impractical to directly fine-tune the SpeechLM on large-scale medical speech corpora. To address this, we first fine-tune the LLM core of the pretrained SpeechLM using large-scale medical text data, and then realign the text and speech modalities with a small amount of medical speech data. This two-stage strategy enables us to transfer general speech capabilities into the medical domain in a data-efficient and scalable manner.
>
> We hope this clarification helps convey the actual intention and contribution of our work. If this explanation makes sense, we would be happy to provide more details and further strengthen the presentation of our method. Although we think there may be some misunderstandings about our work, we are still very grateful for your valuable comments and are more than willing to further improve our work based on them.
>
> > **W.1** \[reviewer misunderstanding\]: Need to clearly articulate the contributions and innovations.
>
> We think the reviewer misunderstanding our work that we just simply train the SpeechLM from scratch using SpeechMedDataset. And we respectfully disagree with the assessment that our methodological design lacks novelty.
>
> We would first like to distinguish our work from the LLaMA-Omni series:
> - **Different Motivation**. The LLaMA-Omni series is a general-purpose SpeechLM using a speech encoder, a speech adaptor, an LLM core, and a speech decoder architecture. Our training strategy is built upon this architecture, and we use LLaMA-Omni2 to validate our idea (rather than using the Qwen model as the base model). Our goal is to develop a universal fine-tuning approach that can efficiently adapt SpeechLMs with similar architectures to vertical domains such as healthcare.
> - **Different Training Strategy**. LLaMA-Omni2, SpeechGPT2 and other SpeechLMs with similar architecture adopt large-scale general speech data to jointly train the adaptor and LLM core in Stage 1, followed by training the speech decoder in Stage 2. In contrast, based on pretrained LLaMA-Omni2 or other SpeechLMs with similar architecture, our two-stage strategy explicitly targets further **domain adaptation**:
>   - **Stage 1 (Knowledge & capability injection)**: We fine-tune only the LLM core from pretrained SpeechLM using large-scale medical text data.
>   - **Stage 2 (Modality realignment)**: We use a small amount of medical speech to re-align the text and speech modality.
>
> And we claim the contributions we made in our work as follows:
> 1. **Training Strategy**. We propose SpeechMedAssist, a medical SpeechLM that introduces speech-based interaction into the medical domain through an efficient two-stage training strategy.
> 2. **Data Construction Pipeline**. We develop a pipeline to convert raw medical dialogues into patient-tailored multi-turn speechbased conversations, creating the first speech medical dialog dataset, SpeechMedDataset.
> 3. **Evaluation Benchmark**. We establish a medical consultation benchmark for SpeechLMs by simulating real medical consultation scenarios, which can demonstrate the efficiency and effectiveness of our approach.
>
> **These represent explorations of applying general-purpose and well-pretrained SpeechLMs to the medical domain, and we believe our work can help advance the application of SpeechLMs to other vertical domains.**

---

> ### Author Response · Authors · 2025-11-26
> **Author Response (2/n)**
>
> > **W.2** \[need explanation, need experiments\]: Should conduct more experiments to compare Qwen2.5-Instruct-7B with the proposed model SpeechMedAssist.
>
> We thank the reviewer for the suggestion to include additional experiments. However, it should be noted that **Qwen2.5-Instruct-7B serves as the base model for LLaMA-Omni2, and our work is built upon LLaMA-Omni2**. Therefore, in our experiments, we primarily compare against LLaMA-Omni2. As shown in our paper, our model achieves significant improvements over LLaMA-Omni2. Nonetheless, we appreciate the reviewer’s suggestions and consider them valuable for further exploration.
>
> - For (a) text-based evaluation, we conducted text-based tests using Qwen2.5-7B-Instruct, and the results are shown below:
>
> | Model               | CMB↑  | CME↑  | MedDG↑ | AIHosipital↑ |
> |---------------------|-------|-------|--------|--------------|
> | Qwen2.5-7B-Instruct | 79.93 | 77.94 | 79.85  | 81.93        |
> | LLaMA-Omni2         | 73.43 | 56.98 | 78.95  | 79.91        |
> | SpeechMedAssist     | 77.96 | 75.48 | 83.29  | 85.52        |
>
> - For (b) speech-based evaluation, we added a baseline with ASR + Qwen2.5-7B-Instruct + TTS, and the results are shown below:
>
> | Model                           | Ency↑  | Safety↓ | MedDG↑ | AIHosipital↑ |
> |---------------------------------|--------|---------|--------|--------------|
> | ASR + Qwen2.5-7B-Instruct + TTS | 60.17  | 1.70    | 80.05  | 80.53        |
> | LLaMA-Omni2                     | 39.82  | 1.96    | 73.18  | 76.33        |
> | SpeechMedAssist                 | 61.02  | 1.32    | 83.26  | 83.40        |
>
> - For (c) text interaction mode, we have presented the performance of all models in **text interaction mode** in Tables 4 and 5 of Appendix B, including our model after the first-stage training.
>
> These additional experiments demonstrate that SpeechMedAssist consistently outperforms both the base LLaMA-Omni2 and Qwen2.5-7B-Instruct in both text and speech interaction settings, highlighting the effectiveness and generality of our two-stage training strategy. We note that on text-based multiple-choice benchmarks, our model performs slightly lower than the Qwen model. This is likely because our training data is primarily dialogue-oriented, which may reduce the model’s performance on tasks requiring multiple-choice reasoning, without affecting its overall medical knowledge or dialogue capability.

---

> ### Author Response · Authors · 2025-11-26
> **Author Response (3/3)**
>
> > **W.3** \[need explanation\]: Why SMA-Stage1 under-performs the original Llama-Omni2 in multi-turn medical question answering.
>
>
> We sincerely appreciate the reviewer’s insightful question. We would like to clarify a potential misunderstanding regarding our **training strategy**. As described earlier, in the first stage, we **trained only the LLM core using a large amount of medical dialogue text data**, while keeping all speech-related components—such as the speech encoder, speech adaptor, and speech decoder—frozen. **As a result, the text and speech subspaces in the embedding space of SMA-stage1 are not fully aligned, which can impact its performance on speech-based tasks**. Consequently, in Table 2, during multi-turn dialogue evaluations based on speech input (MedDG and AIHospital), SMA-stage1 exhibits lower metrics compared to LLaMA-Omni2.
>
> In text-based evaluations presented in Tables 5 (MedDG) and 6 (AIHospital) in Appendix B, SMA-stage1 achieves significantly higher scores than LLaMA-Omni2 and performs comparably to SMA-stage2. This observation supports our explanation and highlights that SMA-stage1 effectively acquires knowledge in the text domain despite the misalignment in speech.
>
> For convenience, we present here the **t2t test results of AIHospital** from the Appendix B in our paper.
>
> | Model       | Symptom Understanding | Active Inquiry | Diagnostic Reasoning | Treatment Advice Validity | Dialogue Quality | Orality Appropriateness | Average |
> |-------------|-----------------------|----------------|----------------------|---------------------------|------------------|-------------------------|---------|
> | LLaMA-Omni2 | 8.08                  | 6.28           | 7.95                 | 8.64                      | 8.80             | 9.15                    | 7.99    |
> | SMA-stage1  | 8.49                  | 7.55           | 8.29                 | 8.54                      | 9.01             | 9.41                    | 8.55    |
> | SMA-stage2  | 8.44                  | 7.57           | 8.21                 | 8.58                      | 8.96             | 9.52                    | 8.55    |
>
> Meanwhile, we also made other interesting observations:
>
> - **Preservation of Text-Domain Knowledge**: In text-based tests such as CMB and CME, SMA-stage1 performs significantly better than LLaMA-Omni2. After the second stage of training with speech data, which realigns text and speech, the CMB and CME metrics remain nearly unchanged, indicating that the knowledge acquired during the text training stage was preserved during speech training.
>
> - **Effective Transfer to Speech Interactions**: The Ency metric evaluates single-turn speech-based medical knowledge Q&A, and the Safety metric evaluates single-turn speech-based medical safety. Although SMA-stage1 was not trained with speech data, it still outperforms LLaMA-Omni2 on these metrics, demonstrating that knowledge acquired in the text space can be effectively utilized during speech interactions.

---

> ### Comment · Reviewer_TkSz · 2025-11-27
> **Response to Authors**
>
> Thanks for your further response. Due to the novelty of this paper, I can not raise my score, and decode to maintain my current score.

---

> > ### Author Response · Authors · 2025-11-27
> > **Response to Reviewer TkSz**
> >
> > Thank you again for your time and for considering our additional clarifications and experimental results.
> >
> > Regarding you mentioned the novelty of our work, we have to reclaim that **we respectfully disagree with the Weakness 1 you raised**. Since we have clearly stated that our work is fundamentally different from the LLaMA-Omni series, we would like to further summarize and clarify the key points.
> >
> > **1. Challenges we faced**
> >
> > - **Limited interactivity in cascaded speech systems**. Conventional ASR–LLM–TTS cascaded pipelines suffer from substantial latency, error propagation, and multimodal information loss, making them difficult to deploy effectively in specialized domains such as healthcare.
> >
> > - **Insufficient speech data in vertical domains**. Most publicly available speech datasets focus on general scenarios. The scarcity of domain-specific speech data makes it challenging to directly fine-tune speech large models for real-world professional applications.
> >
> > **2. Our key innovations**
> >
> > - **A two-stage training strategy**. We first train the LLM core using large-scale medical text data, and subsequently realign the speech–text modalities using a small amount of speech data. To the best of our knowledge, this is the first work to introduce such an efficient and scalable training paradigm.
> >
> > - **A comprehensive evaluation framework for speech LLMs in medical consultation.** We establish a benchmark that constructs a virtual medical consultation environment and assesses both single-turn QA and multi-turn dialogues, combining objective metrics with subjective human evaluations.
> >
> > - **A domain-specific speech data synthesis pipeline.** We design a patient-profile-aware speech dialogue synthesis pipeline tailored to medical consultation scenarios. We also release SpeechMedAssist, which—to the best of our knowledge—is the first speech dataset specifically constructed for the medical domain.
> >
> > **3. Our contributions and broader impact**
> >
> > ```
> > [General-purpose Models]                                      →  [Vertical-domain Applications]
> >
> > Text : LLaMA, Qwen, Deepseek, GLM, InternLM, Mistral...       →  TableGPT, PPTAgent, LawLLM, FinLLM, HuatuoGPT...
> >
> > Audio: SpeechGPT, LLaMA-Omni, Moshi, Qwen-Audio,GLM4-Voice... →  [???]
> > ```
> >
> > The emergence of text-based large models such as BERT and GPT has led to a unified solution framework for numerous NLP tasks, and fine-tuned dialogue LLMs have since been rapidly adopted across many vertical domains.
> >
> > Today, as SpeechLMs such as LLaMA-Omni and SpeechGPT become increasingly prevalent, progress is hindered primarily by the lack of domain-specific speech data.
> >
> > Our work successfully demonstrates the way of general-purpose SpeechLMs to be applied to the medical domain. To the best of our knowledge, we are **the first** to propose a SpeechLM for medical consultation based on end-to-end speech interaction. More importantly, our training strategy, data construction pipeline, and evaluation methodology can be readily extended to other specialized fields such as law and finance. **We believe that our work can advance the application of general-purpose SpeechLMs in numerous vertical domains.**
> >
> > We hope these clarifications provides a clearer understanding of the innovations and contributions of our work, and help address your concerns or misunderstandings about our work. We will sincerely appreciate it if you can re-consider the results.

---

### Official Review · Reviewer_Mkwg · 2025-11-01

**Soundness:** 2
**Presentation:** 3
**Contribution:** 2
**Rating:** 2
**Confidence:** 5

**Summary:**

This paper proposes SpeechMedAssist for enabling speech interaction with patients. To address the challenge of the scarcity of speech data in the medical domain for training Speech LLM for this purpose, this paper proposes a two-stage training paradigm, with the first stage of training the text LLM backbone with rewritten and filtered medical text data to boost the diagnostic and treatment capabilities, and the second stage of speech-text modality re-alignment using synthetic speech dialogue samples that match patient characteristics. Experimental results show that the second stage only requires 10K samples.

**Strengths:**

1.	While the SpeechLLM follows the common architecture, the investigation of decomposing the stage of injecting domain knowledge into the backbone LLM and the stage of speech-text re-alignment for a specific domain, medical consultation, is a valuable investigation and the findings are useful for developing effective domain-specific speech interaction systems. Specifically, the original SpeechLLM is already pre-trained to establish a reasonable speech-text modality alignment. The first stage of knowledge injection only trains the backbone LLM. And the second stage involves training the speech adaptor and LLM with S2T data, and then training the speech decoder with S2S data.

2.	The construction of the TextMedDataset and the SpeechMedDataset is reasonable and clearly presented. The data replay strategy of incorporating the single-turn QA dataset from TextMedDataset in the second training stage helps mitigating catastrophic forgetting of the medical knowledge due to the re-alignment stage.

3.	The analysis in Section 5.5 shows that training directly using medical speech data is more challenging than first helping the core LLM acquiring the domain knowledge and then conducting speech-text re-alignment.

**Weaknesses:**

1.	Some important related open-sourced speech LLMs are missing in the comparisons in Table 2 and Table 3, e.g., Qwen2.5-omni, MiniCPM-o 2.6, Baichuan-omni-1.5, and Step-audio2-mini. This would cause serious problems to comprehensively understand the positioning and effectiveness of the proposed approach.

2.	The first stage of continual training the core LLM with TextMedDataset also suffers from the issue of catastrophic forgetting of the original text and speech capabilities of the core LLM, which is LLaMA-Omni2-7B in this work. It would be very important to evaluate the general text and speech understanding capabilities of the SpeechLLM after Stage I training and Stage II training, using benchmarks such as MMLU for evaluating general text capabilities, and benchmarks such as OpenAudioBench, VoiceBench, Big Ben Audio for evaluating general S2T and S2S capabilities.  Otherwise, it is not clear the text and speech intelligence of the final SpeechLLM.

3.	The evaluation metrics need to be further clarified. Currently, Appendix C explains the 6-dimensions for evaluating multi-turn dialogues, yet it is not clear how they are scored and how the scores of multi-turn dialogues in Table 2 are computed.

4.	For experimental results, it is important to report standard deviation etc to show the stability and help interpret the gains, also
statistical significance tests need to be conducted over the gains.

**Questions:**

Please address the questions raised under Weaknesses.

---

> ### Author Response · Authors · 2025-11-26
> **Author Response (1/n)**
>
> Thank you very much for your valuable and constructive feedback. We sincerely appreciate the reviewer’s careful assessment, and we provide our point-by-point responses below.
>
> > **W.1** \[add baselines\]: Some important related open-sourced speech LLMs are missing in the experiments.
>
> We thank the reviewer for the suggestion. To verify the efficiency of our method and provide a more comprehensive comparison, we add **four additional open-source SpeechLMs** based on your recommendations and tested them across most of our experiments. With these additions, our baselines now cover **three categories of models**:
>
> - **LLM**: HuatuoGPT2, DISC-MedLLM, Zhongjing, Baichuan2-7B
> - **SpeechLLM**: Kimi-Audio, Qwen2-Audio, GLM4-Voice, SpeechGPT2, LLaMA-Omni2, StepAudio2-mini
> - **OmniLMs**: BaichuanOmni-1.5, MiniCPM-o 2.6, Qwen2.5-Omni, ShizhenGPT
>
> The newly added evaluation results are summarized below. We will also update the corresponding tables in the paper accordingly. As shown, SpeechMedAssist achieves superior performance on most metrics, further demonstrating the **effectiveness of our two-stage training strategy**.
>
> | Model            | CMB↑  | CME↑  | Ency↑ | Safety↓ | AIHosipital↑ |
> |------------------|-------|-------|-------|---------|--------------|
> | Qwen2.5-Omni     | 76.83 | 75.33 | 58.12 | 1.72    | 76.53        |
> | BaichuanOmni-1.5 | 64.15 | 70.48 | 62.16 | 1.90    | 80.63        |
> | StepAudio2-mini  | 72.42 | 74.30 | 61.26 | 2.04    | 77.53        |
> | MiniCPM-o 2.6    | 21.68 | 16.01 | 46.45 | 2.08    | 78.60        |
> | SpeechMedAssist  | 77.96 | 75.48 | 61.02 | 1.32    | 83.40        |
>
> > **W.2** \[add benchmark\]: It is necessary to test on general speech datasets to ensure that the general capabilities of the speech large model are not lost during domain adaptation.
>
> We appreciate the reviewer’s concern regarding the preservation of general speech capabilities after domain-specific adaptation. As our work focuses on efficiently applying speech large models to the medical domain, our training data is entirely medical-related, which indeed risks **catastrophic forgetting** of the original text and speech capabilities of the underlying LLM. To address these, we follow the reviewer’s suggestion and evaluate our model on [**MMLU** (https://github.com/hendrycks/test)](https://github.com/hendrycks/test), which is a benchmark for evaluating the text-based QA capability, and [**VoiceBench** (https://github.com/MatthewCYM/VoiceBench)](https://github.com/MatthewCYM/VoiceBench), a comprehensive benchmark covering general reasoning, instruction-following, and speech robustness. The evaluation results are shown in the table below.
>
> | Model                    | MMLU↑ |
> |--------------------------|-------|
> | Zhongjing+ASR+TTS        | 32.81 |
> | Qwen2-Audio              | 51.38 |
> | ShizhenGPT               | 66.36 |
> | GLM4-Voice               | 45.12 |
> | BaichuanOmni-1.5         | 66.25  |
> | LLaMA-Omni2 (base model) | 67.48 |
> | SMA-stage2-10k           | 69.49 |
> | SMA-stage2-198k          | 69.94 |
>
>
> | Model                    | BBH↑  | IFEval↑ | AdvBench↑ | WildVoice↑ | MMSU↑ | AlpacaEval↑ | CommonEval↑ | OpenBookQA↑ |
> |--------------------------|-------|---------|-----------|------------|-------|-------------|-------------|-------------|
> | LLaMA-Omni2 (base model) | 27.13 | 17.31   | 59.80     | 2.538      | 36.26 | 3.584       | 3.118       | 58.13       |
> | SMA-stage2-10k           | 55.81 | 17.18   | 79.80     | 1.692      | 34.72 | 2.358       | 2.028       | 59.80       |
> | SMA-stage2-198k          | 58.14 | 16.49   | 82.69     | 1.705      | 25.83 | 2.380       | 2.046       | 60.66       |
>
>
> From the results, we can see that:
>
> - **General reasoning and instruction-following are preserved**: SMA-stage2 maintains competitive scores on MMLU, BBH, AdvBench, and OpenBookQA, indicating that the model retains strong open-ended reasoning and instruction comprehension despite medical-domain adaptation.
>
> - **Speech robustness and safety remain strong**: Minor drops in WildVoice and MMSU are expected due to domain specialization, but overall the model still handles diverse speech conditions and adversarial instructions effectively, showing that catastrophic forgetting is mitigated.

---

> ### Author Response · Authors · 2025-11-26
> **Author Response (2/2)**
>
> > **W.3** \[need explanation\]: The evaluation metrics need to be further clarified. Currently, Appendix C explains the 6-dimensions for evaluating multi-turn dialogues, yet it is not clear how they are scored and how the scores of multi-turn dialogues in Table 2 are computed.
>
> We apologize for the lack of clarity regarding the multi-turn dialogue evaluation. We will substantially revise Appendix C and the main paper's evaluation section to explicitly detail the scoring process and how the final metrics in Table 2 are computed. And here we make a brief explanation.
>
> - Scoring Mechanism: The "chief examiner" (powered by Qwen2.5-72B-Instruct ) evaluates the dialogue for each of the six dimensions on a discrete scale of 1 to 10. The prompt for the judge explicitly requires the output format to be <Dimension Name>: X/10 - Reason, confirming the 10-point scale. And we visualize these in the Figure 3.
> - Table 2 Computation: The scores reported in Table 2 for both MedDG and AIHospital datasets are the aggregate **average of the scores** across these six dimensions for all evaluated dialogues and multiplied by 10 for easier comparison.
>
>
>
> > **W.4** \[need explanation\]: Should report Standard Deviation and Statistical Significance
>
> We fully agree that reporting measures of variance and statistical significance is critical for robust scientific claims. However, we must address a practical constraint regarding comprehensive reporting of standard deviations:
> - **High Evaluation Cost**: The human-like, LLM-based evaluation process is computationally intensive and time-consuming. Evaluating a single model for one experiment takes several hours, which prevents us from running multiple repetitions and reporting standard deviations for all results across the board.
> - **Observed Stability**: To ensure the accuracy and stability of our key findings, we did conduct multiple-run experiments for our most critical analysis points. We conduct 5 experiments for both step = 5000 and step = 97010 in Figure 4 and include error bars. However, the variance is indeed very small, and the error bars are only visible when zoomed in.
>
> We would like to present the exact means and standard deviations shown in Figure 5 in the table below. As can be seen from the table, after conducting multiple experiments, the variation in the results does not exceed 1%, and therefore we consider the final results to be reliable.
>
> | Model           | Metric     | Mean  | Std Dev  |
> |-----------------|------------|-------|----------|
> | SMA-stage2-10k  | Ency       | 58.14 | 0.4806   |
> | SMA-stage2-10k  | AIHospital | 79.68 | 0.176    |
> | SMA-stage2-198k | Ency       | 61.02 | 0.3663   |
> | SMA-stage2-198k | AIHospital | 83.40 | 0.169    |

---

### Official Review · Reviewer_AsHJ · 2025-11-01

**Soundness:** 3
**Presentation:** 2
**Contribution:** 2
**Rating:** 4
**Confidence:** 4

**Summary:**

This paper presents SpeechMedAssist, a domain-adapted Speech Language Model (SpeechLM) designed to support multi-turn, speech-based medical consultations. The authors propose a two-stage training strategy to overcome the challenge of limited medical speech data: (1) a knowledge and capability injection phase using large-scale rewritten medical text data; and (2) a modality alignment phase using a limited amount (as low as 10k samples) of synthetic speech data.

**Strengths:**

1. The model achieves state-of-the-art results across a range of tasks, outperforming both traditional LLMs with ASR+TTS pipelines and other end-to-end SpeechLMs in both diagnostic performance and speech quality.
2. The two-stage training paradigm is elegant and practical, requiring only ~10k samples to effectively align modalities, a significant reduction in data and compute requirements.
3. Synthetic dataset creation process (with age/gender-aware speech synthesis) is well-considered and likely improves realism and generalizability.
4. The virtual consultation evaluations with roles for patient, doctor, and judge simulate real-world clinical interactions and offer a more grounded benchmark than static Q&A.

**Weaknesses:**

1. Synthetic data realism and risk of overfitting. The paper relies heavily on synthetic speech from TTS models. While they attempt to match speaker attributes (age/gender), it remains unclear how well this generalizes to real patient speech. The paper would benefit from a stronger external evaluation on real-world noisy clinical audio or an ablation comparing synthetic vs. real data alignment.
2. Although the paper evaluates safety via MedSafetyBench and claims improved performance, there is no in-depth error analysis of failure cases or adversarial behavior. Given the high-risk domain, this omission weakens the safety claims.
3. No direct comparison with instruction-fine-tuned SpeechLMs. All comparisons are with general-purpose SpeechLMs or LLMs. The paper should also compare with domain-tuned or instruction-tuned audio models, if available, or at least clarify why they are omitted.
4. Overemphasis on automatic metrics. Metrics like CMB/CMExam are appropriate, but the “vote” metric from clinicians for real-world consultations is presented without sufficient methodological detail (e.g., inter-annotator agreement, voting procedure), reducing its impact.
5. Minor hallucination in output: Figure 1 shows an example where the system incorrectly repeats “apply warm compress to your buttocks,” which is presumably a hallucination or ASR confusion. No systemic analysis of such failures is presented.

**Questions:**

1. How does the model handle real patient audio with varied noise levels and accents? The wild set is small (20 samples). Can you present CER or diagnostic accuracy for this subset or show breakdown by noise level?
2. What mechanisms are in place to prevent hallucinations or unsafe recommendations, especially under low-resource or ambiguous input? Is there a fallback or uncertainty-aware mechanism?
3. Why is the speech decoder untrained in Stage 1 if it contributes to the final generation? Would initializing or co-training help?
4. Did you evaluate the model’s robustness to speech disfluencies (e.g., pauses, fillers, corrections)? This is common in real patients.
5. How scalable is the approach to other domains (e.g., legal, financial) where speech data is also scarce? Do you see any domain-specific challenges?
6. What is the latency distribution across longer conversations? The average latency is good, but variance in multi-turn interaction should be shown.

---

> ### Author Response · Authors · 2025-11-26
> **Author Response (1/n)**
>
> > **Q.1 & W.1** \[need experiments, explanation\]: How does the model handle real patient audio with varied noise levels and accents? The wild set is small (20 samples). Can you present CER or diagnostic accuracy for this subset or show breakdown by noise level?
>
> We thank the reviewer for raising this important question regarding real patient audio and robustness to varied noise conditions.
>
> **For the wild set**, our intention was to address the limitation that both single-turn and multi-turn dialogue experiments rely on synthetically constructed medical consultation scenarios. Although it contains only 20 samples and lacks corresponding ASR labels, these recordings provide a valuable qualitative demonstration of how our model behaves in truly real-world conditions. To facilitate transparent inspection, we have publicly released all 20 audio samples in our anonymous GitHub repository [https://anonymous.4open.science/r/SpeechMedAssist-Anonymous](https://anonymous.4open.science/r/SpeechMedAssist-Anonymous)  and provided an HTML viewer for convenient playback, which can be accessed by simply downloading the repository.
>
> Because this dataset is collected from realistic clinical environments, it contains substantial environmental noise and overlapping speech. Such conditions make manual annotation extremely difficult, and as a result, computing a reliable CER is infeasible. Instead, we adopt expert voting to evaluate the medical quality of the system’s responses, which better reflects how doctors assess interactions in real practice.
>
>
> **For varied noise**, to objectively assess the model’s robustness under different noisy conditions, we conduct controlled single-turn dialogue experiments. Specifically, We randomly sample babble noise and other noise types from the standard noise dataset [\[Microsoft Scalable Noisy Speech Dataset\]](https://github.com/microsoft/MS-SNSD), inject them into the synthesized patient speech at controlled intensity levels (measured by CER with an ASR model), and report the performance of different models under these noise conditions.
>
> Across the noise conditions:
> - All models exhibit expected degradation as noise increases.
> - Our model maintains strong robustness under mild noise (Noise = 0.2).
> - Even under higher noise (Noise = 0.6), SpeechMedAssist remains competitive and outperforms most baselines, demonstrating its effectiveness under varied and challenging audio conditions.
>
> | Noise intensity→      | Noise=0 (CER=9.77%) | Noise=0.2 (CER=10.20%) | Noise=0.6 (CER=12.19%) |
> |-----------------------|---------------------|------------------------|------------------------|
> | Zhongjing+ASR+TTS     | 54.63               | 53.49                  | 50.95                  |
> | ShizhenGPT            | 53.72               | 52.27                  | 49.20                  |
> | Qwen2-Audio           | 49.48               | 46.34                  | 43.85                  |
> | GLM4-Voice            | 54.43               | 54.60                  | 48.25                  |
> | SpeechGPT2            | 56.65               | 54.76                  | 40.68                  |
> | BaichuanOmni-1.5      | 62.16               | 59.15                  | 55.34                  |
> | LLaMA-Omni2           | 39.82               | 30.47                  | 29.78                  |
> | SpeechMedAssist(Ours) | 61.02               | 58.99                  | 58.67                  |

---

> ### Author Response · Authors · 2025-11-26
> **Author Response (2/n)**
>
> > **Q.2** \[need explanation\]: What mechanisms are in place to prevent hallucinations or unsafe recommendations, especially under low-resource or ambiguous input? Is there a fallback or uncertainty-aware mechanism?
>
> Due to the particular nature of the medical domain, we, like the reviewer, place great emphasis on the safety performance of the model, especially in handling hallucinations and avoiding unsafe recommendations. In particular, we mainly adopt two approaches:
>
> - **Intrinsic proactive questioning**. To handle low-resource settings and ambiguous inputs, the model is equipped to proactively ask follow-up questions, as detailed in Section 4.1 and further tested in Multi-turn Conversation. This helps it determine whether the current information is sufficient, request clarification when necessary, and integrate the dialogue history to provide accurate diagnoses and treatment recommendations.
>
> - **External control interventions**. In practical deployment, we use an ASR model to evaluate the clarity of user speech. If the ASR confidence is below 0.75, the model is forced to refuse to answer, preventing potential errors caused by misrecognition. This aspect is more on the engineering side, and we did not mention it in the paper.
>
> We have also conducted extensive experiments to validate the model’s safety performance:
>
> - We use MedSafetyBench, a benchmark specifically designed to evaluate a model’s ability to avoid providing unsafe recommendations. As shown in Table 2 of the paper, our model performs very well.
>
> - We evaluate the model’s ability to proactively ask follow-up questions when information is incomplete in multi-turn dialogue tests. From Figure 3, we can see that the model performs excellently.
>
> - Additionally, we conducted evaluations on a general speech safety benchmark, which is a subset of [VoiceBench](https://github.com/MatthewCYM/VoiceBench) and uses GPT-4o as the judge. The results are as follows:
>
> | Model                    | AdvBench↑   |
> |--------------------------|-------------|
> | LLaMA-Omni2 (base model) | 59.80       |
> | SMA-stage2-10k           | 79.80       |
> | SMA-stage2-198k          | 82.69       |
>
> > **Q.3** \[reviewer's misunderstanding\]: Why is the speech decoder untrained in Stage 1 if it contributes to the final generation? Would initializing or co-training help?
>
> We thank the reviewer for the question and the careful reading. However, we believe there is a misunderstanding regarding our training methodology in comparison to LLaMA-Omni2.
>
> As described in Model Configuration (Section 5.1), we use LLaMA-Omni2 as the base model. Therefore, the speech decoder is already pretrained and effectively initialized from LLaMA-Omni2. In our two-stage training procedure:
>
> - **Stage 1 (Knowledge & Capability Injection)**: We fine-tune the LLM core using large-scale medical text data to inject domain knowledge and capabilities.
> - **Stage 2 (Modality Alignment)**: We train the speech adaptor and speech decoder with a small amount of medical speech data to bridge the modality gap introduced in Stage 1.
>
> This design allows the speech decoder to retain its pretrained generation capabilities while efficiently adapting the model to the medical domain. Co-training the speech decoder in Stage 1 is unnecessary because its parameters are already well-initialized from the pretrained model.
>
>
> > **Q.4** & W.1 \[need explanation\]: Did you evaluate the model’s robustness to speech disfluencies (e.g., pauses, fillers, corrections)? It remains unclear how well this generalizes to real patient speech.
>
> We thank the reviewer for the insightful question. To approximate real patient speech during evaluation, we have taken several measures:
> - **Speech content simulation**: We design detailed prompts and provide authentic patient profiles, including speaking style. For example, a patient profile may specify: “You speak plainly and may not understand complex medical terms, but you can clearly express your feelings and concerns. You would describe an ultrasound finding as ‘there is something in the uterus’ rather than saying ‘an abnormal echo mass in the uterine cavity.’”
> - **Paralinguistic variation**: We use **FishSpeech** to synthesize patient voices, which introduces variability in voice timbre, speaking rate, and other characteristics, simulating natural disfluencies.
>
> Despite these efforts, we acknowledge that fully simulating real patients remains a highly challenging and open problem. We note that some recent studies are dedicated to more realistic patient simulation \[1\], and once such resources are fully publicly available, we are eager to incorporate them to further validate our work.
>
> \[1\] Liu et al., Exploring the Inquiry-Diagnosis Relationship with Advanced Patient Simulators. (2025)

---

> ### Author Response · Authors · 2025-11-26
> **Author Response (3/n)**
>
> > **Q.5** \[need explanation\]: How scalable is the approach to other domains (e.g., legal, financial) where speech data is also scarce? Do you see any domain-specific challenges?
>
> We thank the reviewer for this question. Our core goal is to address the scarcity of speech data in vertical domains and to efficiently apply SpeechLMs to these domains. While we validate our approach in the medical field, the **two-stage training strategy**, comprising Knowledge & Capability Injection and Modality Alignment, is generally applicable to other vertical domains. Similarly, the **data construction pipeline** we designed, presented in Figure 2, is also applicable to other vertical domains. It primarily requires replacing the corpus with domain-specific data and redesigning the data-rewriting prompts to reflect the characteristics of the target domain.
>
> However, for scenarios such as financial or legal consulting, domain knowledge evolves much faster than in the medical field, so it may be necessary to integrate techniques such as **RAG (Retrieval-Augmented Generation)** to supplement real-time or frequently updated information.
>
> > **Q.6** \[need experiments\]: What is the latency distribution across longer conversations? The average latency is good, but variance in multi-turn interaction should be shown.
>
> In the paper, we have reported the response latency of each model for single-turn user speech input. To further verify whether this low latency remains stable as the conversation length increases, we randomly selected 5 dialogue sets and conducted 5 repeated experiments for each. The resulting average response latency as a function of dialogue history length (counting both user query and model response as one turn) is shown below.
>
> | History Length         | 1              | 3              | 5              | 7              | 14                |
> |------------------------|----------------|----------------|----------------|----------------|-------------------|
> | Average Latency (ms)   | **328.4**±36.5 | **349.3**±48.3 | **396.3**±58.6 | **449.7**±77.3 | **748.9.7**±123.3 |
>
> As shown, although the response latency gradually increases with the number of dialogue turns, the growth is relatively small and **remains within an acceptable range**, demonstrating that our system maintains **stable performance in multi-turn interactions**.
>
>
>
>
> > **W.2** \[need analysis\]: Need in-depth error analysis of failure cases or adversarial behavior to strengthen safety claims.
>
> We thank the reviewer for the suggestion. Our model performs very well on MedSafetyBench, with few failure cases. We agree that these cases should be analyzed individually to further enhance the model’s safety, and we will include a statistical analysis of the failure cases in the appendix.
>
> Regarding additional safety strategies, we mainly adopt two approaches:
>
> - **Intrinsic proactive questioning**. To handle low-resource settings and ambiguous inputs, the model is equipped to proactively ask follow-up questions, as detailed in Section 4.1 and further tested in Multi-turn Conversation. This helps it determine whether the current information is sufficient, request clarification when necessary, and integrate the dialogue history to provide accurate diagnoses and treatment recommendations.
>
> - **External control interventions**. In practical deployment, we use an ASR model to evaluate the clarity of user speech. If the ASR confidence is below 0.75, the model is forced to refuse to answer, preventing potential errors caused by misrecognition. This aspect is more on the engineering side, and we did not mention it in the paper.
>
> > **W.3** \[reviewer's misunderstanding\]: Should also compare with domain-tuned or instruction-tuned audio models
>
> We indeed conducted such comparisons, as described in **Section 5.1 (Baselines)**. Specifically, we compared our model with **ShizhenGPT**, a medical multimodal large model that supports speech and image inputs but produces only text outputs.
>
> To further enrich the baseline, we also included **BaichuanOmni-1.5**, a general-purpose speech model that has been trained on a substantial amount of medical data. The evaluation results are summarized in the table below:
>
> | Model            | CMB↑    | CME↑  | Ency↑   | Safety↓   | AIHosipital↑   |
> |------------------|---------|-------|---------|-----------|----------------|
> | ShizhenGPT       | 74.58   | 71.95 | 53.72   | 2.18      | 76.51          |
> | BaichuanOmni-1.5 | 64.15   | 70.48 | 62.16   | 1.90      | 80.63          |
> | SpeechMedAssist  | 77.96   | 75.48 | 61.02   | 1.32      | 83.40          |
>
> As shown, although SpeechMedAssist is trained on less data than existing medical multimodal large models, it achieves superior performance on most evaluation metrics through an efficient two-stage training strategy.

---

> ### Author Response · Authors · 2025-11-26
> **Author Response (4/4)**
>
> > **W.4** \[need explanation\]: Overemphasis on automatic metrics. Metrics like CMB/CMExam are appropriate, but the “vote” metric from clinicians for real-world consultations is presented without sufficient methodological detail (e.g., inter-annotator agreement, voting procedure), reducing its impact.
>
>
> We thank the reviewer for the insightful comment. We have made every effort to adopt objective metrics to evaluate the capabilities of our model. In addition to CMB and CMExam, which assess medical knowledge, we also used objective speech-related metrics such as UTMOS and latency to measure the model’s speech performance. The remaining experiments were conducted based on the following two considerations.
>
> - To accurately and fairly evaluate the model's ability in medical consultation scenarios, which are hard to be evaluated by objective metrics, we constructed virtual multi-turn medical consultation dialogues by following the settings used in works such as AIHospital\[1\] and Baichuan-M2\[2\], and conducted multi-dimensional evaluations across several models.
>
> - To avoid deviating from real medical consultation scenarios, we further evaluated the models with physician voting. Physicians were provided with the same evaluation criteria used in the multi-turn dialogue assessment (detailed in the Appendix C), along with a simple instruction: “Select the one that sounds most like a real doctor.” This procedure was designed to minimize bias and interference, while maintaining fairness and consistency across evaluations.
>
> Through this combination of **objective metrics** and **clinician-guided evaluation**, we aim to provide a comprehensive and reliable assessment of the model’s capabilities.
>
> \[1\] Fan et al., AI Hospital: Benchmarking Large Language Models in a Multi-agent Medical Interaction Simulator. (2024)
>
> \[2\] Baichuan Team, Baichuan-M2: Beyond the Model: Scaling Medical Capability with a Large Verifier System. (2025)
>
>
>
> > **W.5** \[reviewer's misunderstanding\]: Minor hallucination in output: Figure 1 shows an example where the system incorrectly repeats “apply warm compress to your buttocks,” which is presumably a hallucination or ASR confusion. No systemic analysis of such failures is presented.
>
> We sincerely thank the reviewer for carefully reviewing our paper. In Figure 1, we present medical consultation scenarios across three different systems. The repeated phrase highlighted by the reviewer appears in the **cascaded ASR + LLM + TTS system**.
>
> Specifically, the phrase “apply warm compress to your buttocks” appears twice because the **first instance** corresponds to the text output generated by the LLM, while the **second instance** corresponds to the speech output produced by the TTS module, which, when transcribed, yields the same text. Therefore, this repetition is not a hallucination.
>
> To avoid further confusion, we will revise this illustration in the paper to make the distinction between LLM output and TTS output clearer.

---

### Official Review · Reviewer_T5o3 · 2025-11-03

**Soundness:** 2
**Presentation:** 3
**Contribution:** 3
**Rating:** 4
**Confidence:** 3

**Summary:**

The paper proposes SpeechMedAssist (SMA), an end-to-end SpeechLM for multi-turn medical consultations. It adopts a two-stage paradigm: (1) Stage-1 injects medical knowledge and reasoning ability by fine-tuning the LLM core with rewritten and filtered medical text; (2) Stage-2 performs speech–text alignment using a relatively small amount of synthetic medical speech to transfer Stage-1 capabilities into speech interaction. The authors claim that about 10k speech-dialog samples suffice for effective alignment, and build TextMedDataset and SpeechMedDataset along with a multi-dimensional benchmark. Results show SMA performs competitively on single/multi-turn tasks, Wild recordings, speech quality, and latency.

**Strengths:**

- Decoupling knowledge/ability learning from speech alignment, with a clear engineering and theoretical narrative; TTS voices are matched to patient demographics to reduce distribution shift.
- Comprehensive evaluations: knowledge QA, multi-turn dialogue, Wild noisy recordings, speech quality, and latency; comparisons with medical LLMs, ASR/TTS pipelines, and general SpeechLM baselines.
- Clear descriptions of modules (encoder, adapter, LLM core, speech decoder/vocoder) and evaluation metrics.
- Achieving alignment with about 10k speech samples lowers deployment barriers; the model shows advantages in multi-turn and Wild settings, pending stricter external validation.

**Weaknesses:**

1. Results may unfairly favor Qwen-style responses over other families (Llama/Mistral/GLM), as data rewriting, filtering, simulated patients, and multi-turn judges rely on Qwen-family models.

3. Synthetic-to-real gap remains significant. Stage-2 uses only TTS speech, lacking natural hesitation, coughing, pain, accent variation, and complex noise.   Wild experiment is small and subjective. Only 20 clinic recordings with 5 expert votes; lacks objective structured metrics such as key-point coverage and prescription legality.



4. LLM-as-a-judge risk of self-enhancement bias: Qwen2.5-72B is used as judge, while the evaluated model uses Qwen2.5-7B as core.

5. Generality and portability not verified: Claims of general applicability are only demonstrated on Qwen2.5-7B.

6. MedSafetyBench scores are reported, but failure cases (dangerous advice, hallucination, misdiagnosis) are not deeply analyzed.  Lacks discussion of refusal and referral strategies, high-risk intent detection, and confidence calibration.

9. Missing or scattered details on prompts, rejection rules, TTS selection, speaking-rate distribution, noise injection, and parameter configs.

**Questions:**

1. Can the authors provide cross-evaluation using judges and patient simulators from other model families to verify robustness against bias?
2. Is the 10k threshold robust across accents, emotions, speaking rates, and noise? Can multiple seeds and statistical tests be reported?
3. During online dialogues, are there rule-based or knowledge-base checks before giving medical advice? Any detection of overconfidence or hallucinations with fallback to refusal?

**Details Of Ethics Concerns:**

the model gives diagnostic and medication suggestions without a clear safety gateway, refusal strategy, or high-risk filtering.

---

> ### Author Response · Authors · 2025-11-26
> **Author Response (1/n)**
>
> > **Q.1 & W.1 & W.3** \[need experiments\]: The data processing and evaluation pipelines rely exclusively on Qwen models, which may introduce self-enhancement bias. It is necessary to incorporate models from other families for both patient simulation and evaluation.
>
> We sincerely thank the reviewer for the insightful comment. For convenience and reproducibility, we indeed used Qwen models extensively in our pipeline. To address the potential self-enhancement bias, we additionally conducted experiments under three settings: (1) **Simulator & Judge: Qwen2.5-72B (original)**, (2) **Simulator: Qwen2.5-72B, Judge: LLaMA3-70B**, and (3) **Simulator & Judge: LLaMA3-70B**, using the same scoring prompt across all settings. The average results on AIHospital (including newly added models suggested by Reviewer Mkwg) are reported in the table below.
>
> Although all scores decrease when LLaMA3 is used as the simulator or judge, the relative ranking remains highly consistent, indicating that **the benchmark is robust to model family choices** and mitigates the risk of self-enhancement bias.
>
>
> | Model            | Simulator & Judge: Qwen2.5-72B | Simulator: Qwen2.5-72B, Judge: LLaMA3-70B | Simulator & Judge: LLaMA3-70B |
> |------------------|--------------------------------|-------------------------------------------|-------------------------------|
> | HuatuoGPT2       | 80.70                          | 78.83                                     | 77.61                         |
> | Zhongjing        | 77.90                          | 74.93                                     | 72.85                         |
> | Baichuan2-7B     | 72.50                          | 66.96                                     | 66.80                         |
> | ShizhenGPT       | 76.51                          | 72.13                                     | 77.91                         |
> | KimiAudio        | 80.81                          | 76.40                                     | 74.43                         |
> | Qwen2-Audio      | 79.81                          | 73.45                                     | 70.56                         |
> | GLM4-Voice       | 80.43                          | 77.28                                     | 77.31                         |
> | SpeechGPT2       | 80.28                          | 75.98                                     | 68.73                         |
> | MiniCPMo26       | 78.60                          | 74.76                                     | 67.18                         |
> | BaichuanOmni-1.5 | 80.63                          | 77.11                                     | 78.11                         |
> | Qwen2.5Omni      | 76.53                          | 72.01                                     | 70.21                         |
> | StepAudio2Mini   | 77.53                          | 76.45                                     | 74.75                         |
> | LLaMA-Omni2      | 76.33                          | 74.06                                     | 67.41                         |
> | SpeechMedAssist  | 83.40                          | 79.20                                     | 77.70                         |
>
>
> To further validate evaluation reliability, we conducted pairwise comparisons between several high-performing models and ours using GPT-4o as the judge. ShizhenGPT, and BaichuanOmni-1.5 are strong medical-related multimodal models, while GLM4-Voice and StepAudio2Mini are SpeechLMs. Results are summarized in the table below.
>
> | Patient simulator: Qwen2.5-72B, Judge: GPT-4o | Win  | Lose | Tie  | Patient simulator: LLaMA3-70B, Judge: GPT-4o | Win  | Lose | Tie  |
> |----------------------------|------|------|------|-------------------------|------|------|------|
> | SMA vs. HuatuoGPT2         | 53.0 | 41.0 | 6.0  | SMA vs. HuatuoGPT2      | 41.9 | 55.6 | 2.5  |
> | SMA vs. ShizhenGPT         | 83.8 | 16.2 | 0    | SMA vs. ShizhenGPT      | 85.5 | 13.7 | 0.8  |
> | SMA vs. GLM4-Voice         | 53.0 | 33.3 | 13.7 | SMA vs. GLM4-Voice      | 47.9 | 40.2 | 12.0 |
> | SMA vs. BaichuanOmni-1.5   | 56.4 | 38.5 | 5.1  | SMA vs. BaichuanOmni-1.5 | 42.5 | 50.4 | 7.1  |
> | SMA vs. StepAudio2Mini     | 62.4 | 30.8 | 6.8  | SMA vs. StepAudio2Mini  | 52.1 | 42.7 | 5.1  |
>
> When Qwen is used as the patient simulator, our model surpasses all baselines. With LLaMA as the simulator, our method still performs on par or better, confirming the stability of our performance advantage across simulator families.
>
> We have to highlight the data efficiency of our approach: our total speech data (198k fine-tuning samples) is far smaller than that used by other SpeechLMs (e.g., GLM4-Voice ~900k hours), and our medical data (405k samples) is significantly less than that of other medical models.
>
> In summary, all of these demonstrate that our evaluation is fair and robust, and that our model achieves strong performance with substantially less data, fully supporting the conclusions of our paper.

---

> ### Author Response · Authors · 2025-11-26
> **Author Response (2/n)**
>
> > **Q.2 & W.2** \[need experiments, explanation\]: Can multiple seeds and statistical tests be reported? Is the 10k threshold robust across accents, emotions, speaking rates, and noise?
>
> We thank the reviewer for the thoughtful suggestion regarding multi-seed experiments and statistical analysis. However, evaluating a single model under one experimental configuration takes approximately 1–2 hours, which makes extensive multi-seed testing computationally impractical. Nevertheless, to ensure the reliability of the 10k threshold, we conducted **5 independent runs** for both step = 5000 and step = 97010 in Figure 4 and included error bars. The variance across runs is extremely small, and the error bars are only visible when the figure is zoomed in. Given the negligible variance, we did not further invest substantial time and computational resources into running additional multi-seed experiments or statistical significance tests.
>
> For **accents, emotions, and speaking rates**, we have already incorporated these factors into our evaluation. For nearly all test samples, we use **FishSpeech** to randomly sample a reference audio segment from its built-in pool for patient-speech synthesis. This pool contains a broad range of timbres, accents, speaking rates, and emotions, ensuring that the evaluation covers diverse paralinguistic conditions. Under this setup, our conclusions, including the stability of the 10k threshold, remain robust to the variations the reviewer mentioned.
>
> As for **noise robustness**, our training objective is to efficiently adapt pretrained large speech models to vertical domains while using minimal domain-specific speech data. Therefore, we do not augment the limited medical speech data with artificial noise, but instead rely on the inherent robustness of the pretrained speech model. To verify that our fine-tuning strategy does not degrade this robustness, we additionally evaluate the model under noisy conditions in single-turn dialogue tests. We randomly sample babble noise and other noise types from a standard noise dataset [\[Microsoft Scalable Noisy Speech Dataset\]](https://github.com/microsoft/MS-SNSD), inject them into the synthesized patient speech at controlled intensities, and report the results as shown below.
>
>
> | Noise intensity→  | Noise=0 (CER=9.77%) | Noise=0.2 (CER=10.20%)  | Noise=0.6 (CER=12.19%)   |
> |-------------------|---------------------|-------------------------|--------------------------|
> | Zhongjing+ASR+TTS | 54.63               | 53.49                   | 50.95                    |
> | ShizhenGPT        | 53.72               | 52.27                   | 49.20                    |
> | Qwen2-Audio       | 49.48               | 46.34                   | 43.85                    |
> | GLM4-Voice        | 54.43               | 54.60                   | 48.25                    |
> | SpeechGPT2        | 56.65               | 54.76                   | 40.68                    |
> | BaichuanOmni-1.5  | 62.16               | 59.15                   | 55.34                    |
> | LLaMA-Omni2       | 39.82               | 30.47                   | 29.78                    |
> | SMA-stage2-10k    | 58.14               | 55.82                   | 51.79                    |
> | SMA-stage2-198k   | 61.02               | 58.99                   | 58.67                    |
>
> From the results, we observe that all models degrade as noise intensity increases. Our 10k-trained model shows strong robustness under mild noise (Noise = 0.2), performing close to the 198k-trained model. Under higher noise (Noise = 0.6), the 10k model drops more noticeably, while the 198k model remains stable. This indicates that the **10k threshold is sufficient for mild noise**, and robustness under severe noise can be further improved with **simple data augmentation on the 10k samples**.

---

> ### Author Response · Authors · 2025-11-26
> **Author Response (3/n)**
>
> > **Q.3 & W.5** \[need analysis\]: Need analysis of MedSafetyBench failure cases, and it is unclear whether refusal or fallback mechanisms are used to mitigate hallucinations during online diagnosis.
>
> Our model performs very well on MedSafetyBench, with few failure cases. We agree that analyzing these cases is essential for further enhancing safety. Therefore, we will provide a failure-case breakdown and statistical analysis in the appendix.
>
> Regarding additional safety strategies, we mainly adopt two approaches:
>
> - **Intrinsic safety: Proactive clarification and follow-up questioning**. To handle low-resource settings and ambiguous inputs, the model is equipped to proactively ask follow-up questions, as detailed in Section 4.1 and further tested in Multi-turn Conversation. This helps it determine whether the current information is sufficient, request clarification when necessary, and integrate the dialogue history to provide accurate diagnoses and treatment recommendations.
>
> - **External safety: Engineering-level refusal based on ASR confidence**. In practical deployment, we use an ASR model to evaluate the clarity of user speech. if the ASR confidence score falls below 0.75, the system automatically triggers a forced refusal, preventing the model from responding based on potentially misrecognized content. This aspect is more on the engineering side, and we will include a description of this in the appendix.
>
> In the multi-turn dialogue evaluation, we further calculated the **frequency** with which each model proactively asked follow-up questions during its responses (number of inquiries / number of turns), as well as the **quality** of these inquiries. As shown in the table, our model achieves both the highest inquiry frequency and the highest inquiry quality. These results show that our model is significantly more likely to initiate targeted follow-up questions and does so with higher quality, which indicates that it is better able to detect missing information and avoid premature or hallucinated diagnoses.
>
> | Models                | Frequency of proactive inquiries | Quality of follow-up questions |
> |-----------------------|----------------------------------|--------------------------------|
> | Zhongjing+ASR+TTS     | 0.01                             | 6.15                           |
> | HuatuoGPT2+ASR+TTS    | 0.01                             | 6.99                           |
> | DISC-MedLLM+ASR+TTS   | 0.06                             | 7.06                           |
> | ShizhenGPT            | 0.01                             | 7.06                           |
> | Baichuan2             | 0.01                             | 5.34                           |
> | Qwen2-Audio           | 0.02                             | 6.55                           |
> | GLM4-Voice            | 0.06                             | 6.50                           |
> | SpeechGPT2            | 0.37                             | 6.74                           |
> | StepAudio2Mini        | 0.06                             | 6.40                           |
> | BaichuanOmni-1.5      | 0.05                             | 6.65                           |
> | LLaMA-Omni2           | 0.01                             | 5.87                           |
> | SpeechMedAssist(Ours) | 0.25                             | 7.17                           |

---

> ### Author Response · Authors · 2025-11-26
> **Author Response (4/4)**
>
> > **W.4** \[need experiment\]: Generality and portability not verified: Claims of general applicability are only demonstrated on Qwen2.5-7B.
>
> We thank the reviewer for raising this concern regarding the generality of our approach. Our method is designed as a general fine-tuning strategy for adapting pretrained SpeechLMs to vertical domains. However, in the original submission, we only validated it on LLaMA-Omni2 (based on Qwen2.5-7B), which may not sufficiently demonstrate portability.
>
> It is important to note that most mainstream SpeechLMs are closed-source or semi–open-source, with many releasing only inference code. The LLaMA-Omni series is also semi–open-source, which makes conducting additional domain adaptation experiments difficult.
>
> To address this limitation, we additionally conduct experiments on the open-source SpeechLM [**OpenS2S**(https://github.com/CASIA-LM/OpenS2S)](https://github.com/CASIA-LM/OpenS2S) (built on Qwen3-7B and using a similar encoder–adaptor–LLM–decoder architecture). We apply exactly the same training data constructed in our work and follow the same two-stage training strategy proposed in the paper. The results are shown below:
>
> | Model                | CMB↑  | CME↑  | Ency↑ | Safety↓ | MedDG↑ | AIHosipital↑ |
> |----------------------|-------|-------|-------|---------|--------|--------------|
> | OpenS2S              | 65.35 | 66.96 | 52.69 | 2.20    | 74.25  | 69.85        |
> | **SMA(OpenS2S)**     | 61.33 | 61.53 | 55.50 | 1.38    | 80.32  | 79.13        |
> | LLaMA-Omni2          | 73.43 | 56.98 | 39.82 | 1.96    | 73.18  | 76.33        |
> | **SMA(LLaMA-Omni2)** | 77.96 | 75.48 | 61.02 | 1.32    | 83.26  | 83.40        |
>
> For both OpenS2S and LLaMA-Omni2, we observe consistent and substantial improvements on most evaluation metrics after applying our two-stage strategy. **This provides strong evidence that our method is effective and transferable across different SpeechLMs, supporting its generality**.
>
> We also notice a slight performance decrease for SMA (OpenS2S) on the text-only multiple-choice benchmarks (CMB and CME). We attribute this to the fact that most of our training data is dialogue-oriented, which may temporarily reduce the model’s ability to handle multiple-choice formats. Importantly, the model’s medical knowledge itself remains intact, as reflected by the improved performance on the Ency benchmark.
>
>
> > **W.6** \[need explanation\]: Missing or scattered details on prompts, rejection rules, TTS selection, speaking-rate distribution, noise injection, and parameter configs.
>
> We thank the reviewer for carefully reading our paper and providing detailed suggestions. Regarding the points raised, we would like to provide the following clarifications and additional details:
>
> - **Prompts**: we have shown all prompts used in our work as comprehensively as possible in Appendix F, including those used in data preprocessing, inference, and evaluation. If there is anything we may have missed, we would welcome the reviewer to kindly point it out.
>
> - **Rejection rules**: if this refers to the data filtering rules, we have briefly described the filtering details in Section 4.1.
>
> - **TTS selection**: we have provided a detailed explanation in Section 4.2, where we describe the two TTS models we used and the conditions under which each is applied.
>
> - **Speaking-rate distribution**: during data synthesis we considered common speaking rates, but we did not include extremely slow or extremely fast speech. This can be further improved through data augmentation methods.
>
> - **Noise injection**: we did not apply noise injection or other data augmentation methods to the training data, but we conducted experiments on model robustness to noise in our response for Q.2 & W.2.
>
> - **Parameter settings**: we list the key parameters in Section 5.1 under Model Configuration and Training Details. The remaining parameters, such as weight_decay and warmup_ratio, can be found in our codebase, which has been released at [https://anonymous.4open.science/r/SpeechMedAssist-Anonymous](https://anonymous.4open.science/r/SpeechMedAssist-Anonymous) and will be made public after the review process since our original intention is to promote the application of SpeechLMs across various vertical domains.

---

### Author Response · Authors · 2025-11-26
**Brief Summary and Main Revisions**

Dear AC and Reviewers,

Thank all reviewers for your constructive feedback and for your patience in awaiting our response. In response to weaknesses and questions raised, we have provided additional details and conducted further experiments. Below, we summarize the **main points** raised by the reviewers along with our responses, and the corresponding revisions in the paper (highlighted in **blue**).

**We sincerely hope to further discuss with the reviewers to continue improving our work.**

---

> **1. Evaluation: Mitigating Self-Enhancement Bias & Adding Stronger Baselines**

**Problems**
- Using Qwen-family models for evaluation may introduce **self-enhancement bias**.
- Original baselines were limited, making it hard to fully assess model competitiveness.

**Solution**
- **Independent evaluator:** Re-ran AIHospital evaluations using **LLaMA3-70B-Instruct** to remove Qwen-related bias.
- **Third-party preference:** Added **GPT-4o pairwise preference evaluation** to provide an impartial ranking.
- **Expanded baselines:** Included 4 most recently released omni or audio models to strengthen comparisons.

**Experimental Results**
- The evaluation results from LLaMA3-70B show our model remains **competitive or superior**, confirming results are not evaluator-dependent.
- GPT-4o ranking consistently further favors our model compared with other baselines.
- SpeechMedAssist surpasses all expanded baselines and remains SOTA performances.

**Paper Revision**
- We additionally included four baselines in the main experiments, which are listed in Table 2.
- We also added comparison experiments using different models as patient simulators and GPT-4o as the judge, which are shown in Figure 4.

---

> **2. Hallucination Handling and Safety Mechanisms**

**Problems**
- Patients tend to provide **ambiguous or incomplete input**, increasing hallucination risk.
- Existing models may **produce unsafe clinical advice** under uncertain conditions.

**Solution**
- **Proactive inquiry:** Our model is inherently capable of detecting missing information and proactively asking follow-up questions.
- **ASR-confidence-based refusal:** The system **declines answers when ASR confidence is low** to prevent hallucination.
- **Safety benchmarks:** Evaluated on **MedSafetyBench** (medical domain) and **AdvBench** (general-purpose speech domain) to measure hallucination suppression.

**Experimental Results**
- SpeechMedAssist outperforms baselines on MedSafetyBench, avoiding unsafe recommendations.
- In multi-turn ambiguous dialogues, SMA shows strong proactive-question behavior with low hallucination.
- General-purpose speech safety tests: SMA-10k (79.80) and SMA-198k (82.69) significantly outperform base model LLaMA-Omni2 (59.80).

**Paper Revision**
- We have added an analysis of the poor cases in MedSafetyBench in the Appendix F.
- We have added the general speech safety QA evaluation, AdvBench, in Table 3.

---

> **3. Generality of training strategy, General knowledge capability & Robustness of the model**

**Problems**
- Concerns about whether our training strategy is **specific to LLaMA-Omni2**.
- Two-stage training may cause issue of **catastrophic forgetting**.
- Real-world speech includes **noise, emotion, accent**, etc., which may degrade performance.

**Solution**
- **Model generality:** Applied the same training pipeline to **OpenS2S** (based on Qwen3).
- **Knowledge retention：** We evaluate the general capability of our model on **MMLU** and **VoiceBench**.
- **Noise robustness:** Evaluated performance with **3 different noise levels** using **MS-SNSD** .
- **Real-world validation:** We have **publicly released all 20 real-world samples** along with the responses from each model for reference.

**Experimental Results**
- OpenS2S results show consistent gains across almost all evaluation metrics.
- On most metrics of general benchmarks, our model’s performance is on par with or superior to the base model.
- Under light noise, SMA-10k matches the full model; under heavy noise, SMA-198k remains most stable compared with all baselines.
- One can clearly observe our model’s strong real-world performance through the wild dataset.

**Paper Revision**
- We have added the results of MMLU and VoiceBench in Table 3.
- We have presented the performance changes under different noise intensities in Table 3.

---

### Note · Authors · 2026-01-05

I have read and agree with the venue's withdrawal policy on behalf of myself and my co-authors.